# Synthetic Cell as a Platform for Understanding Membrane-Membrane Interactions

**DOI:** 10.3390/membranes11120912

**Published:** 2021-11-23

**Authors:** Bineet Sharma, Hossein Moghimianavval, Sung-Won Hwang, Allen P. Liu

**Affiliations:** 1Department of Mechanical Engineering, University of Michigan, Ann Arbor, MI 48109, USA; bineet@umich.edu (B.S.); mhossein@umich.edu (H.M.); 2Department of Chemical Engineering, University of Michigan, Ann Arbor, MI 48109, USA; sungwonh@umich.edu; 3Department of Biomedical Engineering, University of Michigan, Ann Arbor, MI 48109, USA; 4Cellular and Molecular Biology Program, University of Michigan, Ann Arbor, MI 48105, USA; 5Department of Biophysics, University of Michigan, Ann Arbor, MI 48105, USA

**Keywords:** lipid bilayer membrane, synthetic cells, membrane proteins, membrane fusion, synthetic cell communications

## Abstract

In the pursuit of understanding life, model membranes made of phospholipids were envisaged decades ago as a platform for the bottom-up study of biological processes. Micron-sized lipid vesicles have gained great acceptance as their bilayer membrane resembles the natural cell membrane. Important biological events involving membranes, such as membrane protein insertion, membrane fusion, and intercellular communication, will be highlighted in this review with recent research updates. We will first review different lipid bilayer platforms used for incorporation of integral membrane proteins and challenges associated with their functional reconstitution. We next discuss different methods for reconstitution of membrane fusion and compare their fusion efficiency. Lastly, we will highlight the importance and challenges of intercellular communication between synthetic cells and synthetic cells-to-natural cells. We will summarize the review by highlighting the challenges and opportunities associated with studying membrane–membrane interactions and possible future research directions.

## 1. Introduction

Nature is the prime source of inspiration for humans to understand life and create something new. Many of today’s technologies have been inspired from our surroundings, for example, the invention of flight (birds), submarine (whales), Shinkansen bullet train (kingfisher), sonar (dolphin and bats), and many more. The scientific community has not been untouched by this inventiveness that has led to many novel research branches, for example biomimicry, synthetic cell research, and in vitro protein synthesis as the frontiers in the field of synthetic biology [1,2,3,4,5,6].

A natural cell was thought to have a simple construction, which later turned out to be a highly complex unit of life hosting many reactions with spatiotemporal precision [7]. Cell membrane is the first boundary that draws the physical existence of the cell. It serves as a barrier where internal materials (DNA, proteins, and organelles) are separated and protected from the outside environment. It also acts as a gatekeeper that allows unassisted passage of substances such as water and gases, but not large molecules. This semi-permeable nature of the cell membrane is regulated by diffusion or with the aid of special transporters across the cell membrane such as ion channels and transporters. The cell membrane plays a vital role in almost all the cellular processes, such as endocytosis, exocytosis, membrane fusion, inter- and intra-cellular communications, and fertilization.

Researchers have been trying to mimic cellular functions to enhance their understanding about vital biological processes. To achieve this, bilayer membranes made with amphiphilic phospholipids are used. Many platforms are available where lipid bilayers are created, including standing bilayer membrane, supported bilayer, and unilamellar lipid vesicles (see Section 2). Standing bilayer is utilized for the study of ion channels and membrane proteins [8], while supported bilayer is utilized for membrane fusion [9] and protein expression [10,11]. Micron-sized unilamellar vesicles are used as a synthetic cell model for studying biochemical reactions in vitro [7,12]. We will highlight some of the extensively studied membrane proteins and recent research updates.

Significant efforts have been made in recapitulating different aspects of membrane–membrane interactions, such as vesicle–vesicle fusion. We will highlight the conditions required to achieve membrane fusion and challenges associated with it. We will primarily discuss DNA-mediated and coiled-coil peptide-mediated fusion and their potential future direction. Lastly, inter/intra-cellular communication will be discussed, where we will highlight some recent research on synthetic cell-to-synthetic cell and synthetic-to-natural cell communications.

## 2. Model Bilayer Membranes

### 2.1. Planar Lipid Bilayer

#### 2.1.1. Black Lipid Membrane

‘Black’ lipid membrane (BLM) received its name because of its appearance by optical microscopy. BLM was discovered some 60 years ago when Mueller et al. [13,14] reconstituted the first cell membrane structure in vitro from extracted brain lipids and measured the electrical polarization across the membrane. It is a unique setup where electrodes are placed at both sides of a standing lipid bilayer and the electrical conductivity across it is measured. Usually, two aqueous chambers are separated by a planar bilayer spanning an aperture on a hydrophobic (i.e., made of Teflon) septum, with its size varying from 50 µm to 1 mm. There are two types of BLM depending on the orientation of the orifice, horizontal and vertical BLMs (Figure 1A). In both the cases, the bilayer is formed by either pseudo-painting or dragging a lipid bubble over the aperture while the chambers are filled with buffers. Typically, for the incorporation of membrane proteins, diphytanoyl phosphatidylcholine (DPhPC) or a mixture of 1-palmitoyl, 2-oleoylphosphatidylethanolamine (POPE) and 1-palmitoyl, 2-oleoylphosphatidylglycerol (POPG) (3:1) [15] are used with concentration varying from 5 to 20 mg/mL in organic solvents such as n-decane, hexadecane, or hexadecane/hexane (10:1).

The main limitation of this method, apart from the fact that the bilayer formed by this method is fragile, is the presence of extra solvent in the bilayer membrane that compromises the measurements. Subsequently, a solvent-free bilayer assembly method was introduced by Montal and Mueller [16]. In this setup, two aqueous chambers with lipid monolayer on their surface are partitioned by a Teflon septum with an aperture above the water surface. The Teflon septum is then lowered gradually into an aqueous bath which subsequently leads to the formation of a bilayer devoid of any solvent. However, this approach does not resolve the membrane stability issue. Over the years, BLM has proven to be a powerful method for studying the electrophysiology of ion channels and membrane proteins. The incorporation of membrane proteins to BLM is usually achieved by direct insertion of purified proteins into the bilayer or incorporated by fusion of proteoliposomes to the standing bilayer membrane. BLM is most useful for probing the biophysical properties of channels. However, if a membrane protein has a conformational change that induces downstream effects, such effects will be difficult to detect in a BLM setup. Hence, utilizing this platform is mostly beneficial when studying the selectivity, conductivity, and drug pharmacology properties of reconstituted proteins.

Supported lipid bilayer (SLB) is a planar bilayer formed as a result of self-assembly of phospholipids on a hydrophilic (i.e., oxidized glass) surface [20,21]. The fluid nature of SLBs is restored by the presence of a thin water layer of ~1–2 nm between the solid surface and the bilayer (Figure 1B) [22,23,24]. Unlike a suspended lipid bilayer in BLM which is quite fragile, SLB provides a robust and stable platform for surface-specific interactions. The stability of SLB is achieved by the efficient interaction between the solid support and the planar bilayer governed by van der Waals, hydration (a repulsion force between two hydrated surfaces), electrostatic, and steric forces [25,26].

There are majorly two different methods used for the formation of SLB: the Langmuir–Blodgett deposition method and the vesicle fusion method. In the former method, the lower leaflet of the monolayer lipid is created at the air–water interface by slowly pulling out the hydrophilic surface submerged in the aqueous phase. The second step of the Langmuir–Blodgett method involves the deposition of a second lipid monolayer leaflet by horizontally dipping the surface to the air–water interface [20]. In the vesicle fusion method, SLB is formed by adsorption and fusion of small unilamellar vesicles (SUVs) on a hydrophilic surface [27,28]. A combination of the above two methods has been employed where a bilayer was formed by vesicle fusion to a solid-supported lipid monolayer [29], and is useful in creating asymmetric bilayers [29,30]. The most-used substrates for SLBs are fused silica [20], mica [31,32], borosilicate glass [20,26], and oxidized silicon [20]. Other thin-film metal surfaces have been employed for supported bilayers, including TiO_2_, indium-tin-oxide, gold, silver, and platinum [33].

#### 2.1.2. Droplet Interface Bilayer

Like any other techniques, the aforementioned methods have limitations, including mechanically unstable bilayer (i.e., BLM), accessibility to only one side of the membrane, and surface defects due to non-uniformity (i.e., SLB). A similar approach of what we know today as the droplet interface bilayer (DIB) was introduced by Tsofina et al. [34] in 1966, almost the same period of time as when BLM and SLB were introduced. DIB is formed when two aqueous droplets with a lipid monolayer on their surface are brought together while submerged in a lipid-oil mixture [35,36]. There exists two ways to achieve DIB, one where lipids are dissolved in oil (e.g., hexadecane or squalene) and the other where lipids are introduced in the form of SUVs inside the aqueous droplet to the water–oil interface [17,37]. They are commonly referred to as lipid-out and lipid-in, respectively (Figure 1C). This platform has been used in electrophysiology studies of membrane/ion channels by either forming droplets on the tip of the agarose/agar-coated electrodes [35,38] or by placing the electrodes at the bottom of the droplets [39]. A modification of DIB was introduced as a droplet on the hydrogel-supported bilayer (DHB) as a more robust method [18,40]. In DHB, an agarose substrate is formed on a glass surface followed by the addition of a lipid-oil mixture, and this step forms the lipid monolayer on the agarose surface. An aqueous droplet hanging on an agarose-coated electrode is submerged in an oil bath, spontaneously forming a monolayer. Then, the aqueous droplet is gently dropped to the agarose surface, where it forms a bilayer. The bilayer formed by this method is stable for weeks and even resistant to mechanical shock (Figure 1D). Due to the defined compartmentalization in this method, DIBs have been used to recapitulate communication pathways and feedback cascades mediated by membrane proteins and porins. The ability to visualize diffusion of fluorescent molecules across droplets as well as to record electrical currents between droplets makes DIBs ideally suited for studying protein-mediated interactions between droplet-in-oil synthetic cells. The downside, however, is that the outer organic phase does not resemble the aqueous solutions in a natural environment, and an oil–water interface may result in protein denaturation.

### 2.2. Vesicle Preparation

Unlike planar lipid bilayers, vesicles have an independent compartment that resembles a natural cell, separating an inner aqueous volume from an outer aqueous environment. Giant unilamellar vesicles (GUVs) are the most widely used model as synthetic cells and as protocells from the origin of life perspective [41]. They have a size between 1 and 100 μm in diameter, which is in the range of biological cells. Giant vesicles made of lipids are fragile, and this has motivated the use of other membrane-forming materials. Polymersomes are comprised of chemically synthesized amphiphilic polymers that self-assemble to form vesicles [42]. They are robust and are considered to be mechanically tough with low permeability for ions as compared to lipid vesicles. Here, we focus our discussion on lipid bilayer giant vesicles as lipids are the natural substrate for membrane proteins. There are several methods available for the preparation of GUVs, and each of the methods is introduced here with its advantages and disadvantages.

#### 2.2.1. Hydration Method

The hydration method, also referred to as spontaneous swelling or gentle hydration, is regarded as one of the first approaches to make GUVs [43,44]. It involves a two-step procedure: (i) dehydration of the lipid on a substrate of choice (mainly glass), followed by (ii) rehydration of the deposited lipid films with a solution to be encapsulated. During the rehydration, the temperature has to be higher than the lipid phase transition temperature to form GUVs [45]. One of the main disadvantages of this method is that the rehydration step requires a long incubation time, ranging from several hours to overnight [46]. An alternative method to accelerate this process is a widely used method called electroformation or electroswelling [47]. By applying an alternating current (AC)-electric field on an electrically conductive surface (i.e., glass coated with indium tin oxide or platinum wires), a high yield of vesicles is achieved in a relatively short time (Figure 1E) [48]. Still, drawbacks exist since the hydration method, including electroformation, cannot readily encapsulate large-sized molecules or be used with high ionic strength solutions (for electroformation). To overcome this, microinjection of molecules through microneedles can be a viable strategy, whereby only a limited amount of volume can be injected [49].

#### 2.2.2. Droplet Microfluidics Method

The main benefit of introducing microfluidics technology in making GUVs is that more uniform vesicles can be generated [50]. By using a microfluidics device, water-oil-water double emulsion droplets are formed with low polydispersity [51]. The middle oil phase is removed sequentially to allow lipid monolayers in water–oil and oil–water interfaces meet and form vesicles with lipid bilayers (Figure 1E). A significant advantage of this method is that it is much more efficient in encapsulating large molecules compared to the hydration method. However, drawbacks also exist because it is difficult to remove the oil phase completely; therefore, there may be residual oil in the vesicle membrane [52]. The presence of oil in the membrane has been a long-standing problem since oil can affect the biophysical properties of the membrane [53]. In this regard, an alternative method that replaces oil with alcohol has been developed [54]. Another well-known method that uses a microfluidics technique is called microfluidic jetting [55,56]. While the vesicles are generated from the planar lipid, some oil is still expected to be present in the membrane [57].

#### 2.2.3. Inverted Emulsion Method

Developed by Pautot et al. [58], this method starts with forming water-in-oil single emulsion droplets with a lipid monolayer using various methods. Droplets are then placed on top of another oil–water interface with a lipid monolayer. Through centrifugation, droplets pass through the interface and lead to the formation of GUVs (Figure 1E). While residual oil may also be present in the membrane, the inverted emulsion method is widely adopted due to its simplicity (i.e., requires little time) and versatility (i.e., little restriction on what can be encapsulated) [59].

#### 2.2.4. cDICE Method

One notable variant of the inverted emulsion method is called continuous droplet interface crossing encapsulation (cDICE), which improved on some of the drawbacks of the inverted emulsion method [60,61,62,63]. Water-in-oil droplets are injected into the custom-designed cylindrical chamber mount on a tabletop centrifuge, where the oil-aqueous phase is formed by the centrifugal force. Instead of pushing all the droplets at once, droplets are pushed one by one through a capillary (Figure 1F) [19]. This allows lipid components to saturate at each interface, thereby forming more stable and high-yield vesicles. However, because the cDICE method also uses oil as a lipid solvent, it is likely to contain some residual oil in the membrane.

Since each of the methods developed to form GUVs presents different pros and cons, it is important to select the proper method depending on the experimental needs. For instance, in cases where the remaining oil in the membrane may have a critical effect, such as examining the properties of membrane proteins, the hydration method can be more appropriate than the others [57,64]. However, if efficient encapsulation of large molecules such as enzymes is important or if uniform vesicle size is required, methods other than hydration should be considered. In addition to choosing the most suitable methods over others, there is plenty of room for the improvements of the existing approaches as well as the development of entirely new techniques that can overcome the shortcomings of the current methods.

## 3. Membrane Protein Incorporation into Lipid Bilayer

The majority of interactions that membranous structures have with either each other or their external environment are mediated by membrane proteins. Using synthetic cells as model membranes for studying membrane–membrane interactions is not possible unless their membrane is decorated with various functional proteins that allow interactions with the outer environment. As GUVs are used primarily as a model for synthetic cells, the ability to reconstitute membrane proteins into their lipid bilayers is an important consideration [65]. Below, we showcase the reconstitution of four different membrane proteins.

### 3.1. Alpha Hemolysin

α-Hemolysin (αHL) is a water-soluble toxin secreted by *Staphylococcus aureus* that targets both prokaryotic and eukaryotic cells [66]. αHL is secreted as a monomer but forms a transmembrane heptameric pore when lodged in the bilayer membrane of the target cell [67], eventually causing cell death due to transport of small ions and low molecular weight molecules [68]. Structurally, αHL forms a β-barrel protein pore of 2.6 nm made of 14-strand of anti-parallel β-sheets from 7 αHL monomers [69]. Inspired by biological nanopores such as αHL, there has been significant advancement in the field of synthetic nanopores, especially ones designed with DNA [70,71]. Unlike αHL, the size of DNA-based nanopores can be tuned from 4 to 30 nm in diameter [72,73,74], and these nanopores have been applied in numerous sensing applications [75].

αHL has been reconstituted using different bilayer platforms, such as liposomes [76,77], SLBs [78,79], and DIBs [39]. The ease of αHL self-assembly in the membrane enables a variety of applications that require transporting ions and molecules across the membrane, including biosensing [39], coacervation [80], and activation of genetic circuits [11]. For example, Adamala et al. generated liposomes encapsulating genetic circuits and cell lysates with transcriptional and translational activity and used αHL to enable membrane permeability of small molecular inducers (Figure 2A) [81]. More recently, Hilburger et al. developed a membrane AND gate where the release of the encapsulated material was dependent on a fatty acid and αHL [82].

### 3.2. Mechanosensitive Channel (MscL)

The bacterial mechanosensitive channel was discovered as a channel that responded to suction during patch-clamp experiments [86]. The mechanosensitive channel of large conductance (MscL) has been extensively studied as one of the model membrane proteins owing to its highly conserved structure and function between bacteria species. In nature, MscL functions as an emergency release valve that prevents cell lysis when bacteria are exposed to severe osmotic downshifts. It consists of five identical subunits that open its pore of ~3 nm diameter when the membrane tension reaches the threshold of 10~12 mN/m [87,88]. As a nonselective channel with the largest known pore size, there is great opportunity to use MscL in building mechanosensitive synthetic cells [89].

Attempts to reconstitute MscL have been made in various types of planer lipid bilayers, including DHBs, SLBs, and DIBs [11,90,91,92]. Among them, Haylock et al. and Strutt et al. demonstrated an indirect trigger of MscL by adding trimethylammonium ethyl methanethiosulfonate (MTSET) or lysophosphatidylcholine (LPC) [91,92]. It has only been in recent years that MscL was used in synthetic cells. Majumder et al. reported the development of mechanosensitive synthetic cells expressing MscL by using cell-free expression (CFE) [83]. The synthetic cells responded to osmotic down-shock and activated a fluorescence calcium reporter (Figure 2B). Following this study, Garamella et al. created synthetic cells capable of responding to osmolarity down-shock and inducing expression of a cytoskeletal protein MreB [93]. In a study carried out by Hindley et al., a vesicle-in-vesicle structure was made where calcium influx was initiated by αHL addition, and it subsequently activates phospholipase A2 and leading to the release of dye molecules through MscL (Figure 2C) [84]. Since MscL has been studied in detail, various MscL mutants have been investigated, such as those that exhibit a lower activation threshold (~6 mN/m), temperature sensitivity, or chemically inducible features [88,94,95]. Other stimuli have also been shown to induce MscL activation, including pH, light, and ultrasound [94,95,96]. Given the tunability of MscL activities and stimuli-responsiveness, it is expected that MscL will continue to be actively deployed in the synthetic cell field.

### 3.3. SUN Proteins

The presence of nuclear envelope (NE) in eukaryotic cells is one of the features that differs between eukaryotic and prokaryotic cells. Cellular functions such as protein synthesis, cell migration, and chromosome dynamics require a definite nuclear positioning which is regulated by LINC complexes (linker of nucleoskeleton and cytoskeleton). LINC complexes are comprised of SUN (Sad1, UNC-84) proteins, located in the inner nuclear membrane (INM), and KASH (Klarsicht, ANC-1, and Syne Homology) proteins in the outer nuclear membrane (ONM) [97]. Both SUN and KASH domains form a bridge between INM and ONM which plays a crucial role in nuclear positioning and transmission of mechanical force across NE during meiosis [98,99].

Our lab recently demonstrated the use of a HeLa-based CFE system for orientation-specific reconstitution of the LINC complexes proteins SUN1 and SUN2 [100]. SUN proteins are located in the NE between INM and ONM, such that they are inaccessible to direct biochemical assays. In this study, we showed that SUN proteins expressed in HeLa CFE reactions inserted into bilayer membranes on supported bilayers with excess membrane reservoir (SUPER) templates. Using a protease protection assay, we determined the topology of SUN1 and SUN2 and discovered an additional transmembrane domain and hydrophobic regions that were previously unidentified. The directional reconstitution of SUN proteins was most likely mediated by microsome fusion to SUPER templates [101]. The utility of a mammalian CFE system for reconstituting complex membrane proteins will open up more opportunities for creating synthetic cells with advanced sensing capabilities.

### 3.4. Bacteriorhodopsin

Bacteriorhodopsin is a seven-pass transmembrane protein from Archaea that drives protons across the membrane using energy from light [102,103,104]. The interest in reconstituting bacteriorhodopsin in membranes aiming to create artificial photosynthetic entities has a long history. First, in the work of Racker and Stoeckenius [105], purple membrane vesicles of *Halobacterium halobium* that contain bacteriorhodopsin were reconstituted and used to catalyze light-dependent ATP production. Later, different strategies were implemented to reconstitute bacteriorhodopsin in the membrane of liposomes or GUVs. For example, a method for detergent-mediated reconstitution of functional bacteriorhodopsin was presented by Dezi et al. [106]. Kahya et al. [107] proposed a method based on peptide-induced fusion to introduce bacteriorhodopsin-containing proteoliposomes into the membrane of GUVs as well. Lastly, detergent-mediated methods that rely on CFE of bacteriorhodopsin were shown to reconstitute functional proteins [108,109].

In order to produce light-driven energy production, bacteriorhodopsin is usually co-reconstituted with ATP synthase subunits F_0_ and F_1_ [110]. Reconstitution of both proteins in polymersomes of amphiphilic triblock copolymer, PEtOz−PDMS−PEtOz (poly(2-ethyl-2-oxazoline)-b-poly(dimethylsiloxane)-b-poly(2-ethyl-2-oxazoline)) has been shown to create nano-scale photosynthetic organelles [111]. In a different study, instead of reconstituting both bacteriorhodopsin and ATP synthase on the same membrane, Chen et al. coated the surface of plasmonic colloidal capsules, made by assembly of Au-Ag nanorods, with the purple membrane of *Halobacterium halobium* containing bacteriorhodopsin [112]. The neighboring nanoparticles of colloidal capsules created concentrated electric fields that caused increased light absorption by bacteriorhodopsin. The improved photo-absorption system was then coupled with proteoliposomes that harbored ATP synthase, creating a complete artificial photosynthetic system. The development of methods to create artificial photosynthetic entities expedited the translation of bacteriorhodopsin into applications in synthetic cells. Recently, artificial photosynthetic organelles were designed by Ahmad et al. [113]. These nanometer-sized organelles were used to activate flagellar motion as well as contraction of microtubule networks by kinesin-1 motors. Through oscillatory light illumination, Ahmad et al. were able to control the flagellar beating frequency. Similarly, proteoliposomes that contained bacteriorhodopsin and F_0_ and F_1_ ATP synthase subunits were used as energy-producing organelles to generate ATP for cell-free protein synthesis inside synthetic cells (Figure 2D) [85]. Finally, designing photosynthetic organelles is not limited to bacteriorhodopsin. For example, Lee et al. have demonstrated energy production by synthetic organelles made of ATP synthase and photoconverters, including plant-derived photosystem II and bacteria-derived proteorhodopsin [114]. The energy produced by these nanometer-sized organelles was then coupled to polymerization of actin filaments.

The significant progress on methods and strategies of reconstituting bacteriorhodopsin on lipid bilayer membranes to create synthetic energy-producing organelles has certainly paved the way to create self-sustaining, long-lived synthetic cells. By coupling light-driven energy production to cell motion, one can envisage more life-like synthetic cells in the near future.

## 4. Membrane Fusion

Membrane fusion involves close contact between two bilayers that eventually leads to a single merged membrane (Figure 3A) [115]. Membrane fusion is a vital process in eukaryotic cells. It regulates major cellular process such as cellular trafficking, exocytosis, fertilization, and endocytosis. The most important conditions for lipid bilayers to fully fuse are the lipid composition [115,116] and the close distance between the two bilayers. There are numerous approaches to promote membrane fusion, including metal ion-induced fusion, DNA-mediated, peptide nucleic acid (PNA)-mediated, and coiled-coil peptides. Readers are referred to the excellent review articles for additional details [116,117]. In the section below, we will focus on DNA-mediated and peptide-mediated fusion and highlight some recent studies.

### 4.1. DNA-Mediated Fusion

DNA-based interaction provides an excellent strategy for membrane fusion due to the high selectivity between DNA strands. In this approach, cholesterol- or lipid-anchored DNA spontaneously becomes part of the membrane, with DNA strands exposed on the outer surface of vesicles [118,119]. By bringing apposing vesicles into close proximity, fusion of the bilayer membrane occurs due to hybridization of DNA strands. Membrane fusion can be confirmed by lipid mixing and content mixing. Later, Hook and co-workers investigated the effect of DNA length, number of DNA strands, and number of cholesterol groups on membrane fusion [120]. When comparing the efficacy of fusion between single-stranded DNA and double-stranded DNA with overhang (complementary overhang on the other vesicle), double-stranded DNA showed improved binding affinity compared to single-stranded DNA, where there was some degree of dissociation of hybridized strands [119,120]. In case of single-stranded DNA and a single cholesterol group, content leakage and dissociation of docked vesicles were observed [120]. They also found that longer DNA strands increased vesicle docking but failed to lead to vesicle fusion. Recently, Peruzzi et al. showed the initiation of CFE by DNA-mediated vesicle fusion and found that phase-segregation of membrane domains enhances fusion between different vesicle populations (Figure 3B) [121]. Controlling fusion by using DNA-tethered vesicles provides exquisite specificity and expands the opportunities to control spatiotemporal dynamics of CFE reactions.

### 4.2. Peptide-Mediated Fusion

There exists numerous demonstrations of peptide-mediated membrane fusion in the past, with examples such as vancomycin glycopeptide and D-Ala-D-Ala dipeptide or peptide nucleic acids [123,124]. Soluble *N*-ethylmaleimide-sensitive factor attachment protein receptors (SNAREs)-mediated fusion has proven to be most efficient and closest to biological systems. SNAREs were identified as the key molecular players mediating membrane fusion [125,126]. There exists more than 30 SNARE family members in mammalian cells. Complementary sets of SNARE proteins, present on respective membranes, form a stable four-coiled-coil α-helix bundle, which ultimately leads to membrane fusion [127].

Inspired by the four-helix bundle formation, Kros’ group has developed a SNARE mimicking system comprised of lipid-conjugated oligopeptides with PEG as a spacer [128]. To mimic the four-helix bundle complex, they introduced three heptad repeat units of lysine-rich and glutamic acid-rich amino acids. These oligopeptides form a stable heterodimer with a dissociation constant of ~10^−7^ M [129]. A recent study, reported by the same group, demonstrated membrane fusion between GUVs with peptide K (KIAALKE)_4_ and LUVs with peptide E (EIAALEK)_4_ (Figure 3C) [122].

## 5. Intercellular Communication

One of the most defining characteristics of natural cells is their ability to sense each other, communicate, and act as a consortium. Quorum-sensing, as an example of intra- and inter-species communication, is an essential aspect in bacteria population growth and a regulator of physiological processes [130,131]. In eukaryotic cells, for example, collective migration of a cohort of cells versus single-cell locomotion highlights the importance of the exchange of mechanical cues and mechano-sensing [132,133,134]. The ability to sense the environmental cues as well as sending signals heavily relies on the existence of proteins residing on the membrane of natural cells. Membrane proteins such as GPCRs are critical in signaling cascades for cells to respond to changes in their environment. The advances in membrane protein reconstitution methods described earlier have led to significant progress in reconstituting intercellular communication among synthetic cells.

### 5.1. Synthetic Cell–Synthetic Cell Communication

For successful biomimicry of natural cells as well as the creation of active materials, it is crucial for synthetic cells not only to be able to sense their environment and the presence of other synthetic cells, but to also communicate with them via signaling molecules. The difficulty of mimicking intercellular communication mechanisms can be attributed to the high complexity and specificity of extracellular signaling molecules and their targeted secretion in natural cells, whereas synthetic cells merely rely on natural diffusion of small molecules based on chemical gradients. Since most synthetic cells are compartmentalized by phospholipid membranes, synthetic cell communication designs exploit the physical and biochemical properties of lipid bilayers, such as their semi-permeability and their ability to host porins such as αHL. For example, gene-mediated communication between synthetic cells was engineered by encapsulating the non-permeable molecule doxycycline (Dox) in one population and a plasmid encoding firefly luciferase (fluc) under a Tet promoter in the other population [81]. The release of Dox from the first community of synthetic cells and the entry of Dox into the second synthetic cell population, both mediated by αHL, triggers the synthesis of fluc. Additionally, further genetic circuits are engineered that rely on free diffusion of Arabinose across liposome bilayer membranes or depend on SNARE-mediated fusion of two different populations of liposomes. A drawback of such a system is that signaling heavily relies on one molecule and its natural diffusion rate, leading to inefficient signal propagation that fades over time.

To overcome the aforementioned drawback, Buddingh et al. designed sender synthetic cells that use adenosine monophosphate (AMP) as the signaling molecule [135]. Upon diffusing into the receiver cells through αHL, AMP binds to glycogen phosphorylase b and allosterically activates the enzyme, which leads to the production of NADH through a cascade of reactions. This signal amplification strategy allows the system to propagate the signal over long distances as one molecule of AMP activates an enzyme that produces a large amount of NADH (Figure 4A).

In an uncommon approach, the membrane has been used as a part of the signaling cascade, where phospholipid vesicles are sender cells and proteinosomes displaying an enzyme are receivers [139]. The two populations of synthetic cells communicate using glucose as the signal molecule and the receiver synthetic cells process glucose via glucose oxidase (GOx) as a component of their membranes.

In another work, Yang et al. demonstrated a DNA-origami-based pore that opens only when two synthetic cells are in contact, allowing material exchange only when two synthetic cells are in close proximity [140]. Such a design can significantly help in concentrated signal release, in contrast to the uniform release of molecules through αHL. Another innovative example of concentrated signal release upon synthetic cells’ contact is the work of Chakraborty et al., where the synthetic cell adhesion between prey and predator populations is triggered upon bioluminescence from predator cells that, in turn, dimerizes proteins iLID and Nano, each of which resides in the membrane of one group of synthetic cells [141]. The dimerization reconstitutes synthetic cell adhesion, which leads to opening of αHL that is blocked unless synthetic cells are in contact [142]. The opening of αHL activates phospholipase A_2_ (PLA2) inside the prey cells through diffusion of calcium ions from predator cells. Activation of PLA2 causes the cleavage of phospholipid acyl chains that leads to the collapse of prey cells. Lastly, quorum-sensing of synthetic cells has been shown by Niederholtmeyer et al. [136]. In their work, the synthetic cell’s membrane is composed of porous polymer acrylate that allows diffusion of molecules up to 2 MDa. Due to this diffusion constraint, receiver synthetic cells produce desirable signals based on their distance from the sender cells and only when their population is above a certain density (Figure 4B).

In addition to compartmentalized synthetic cells, synthetic cell communication has been reconstituted between water-in-oil droplets as well as liquid–liquid phase-separated droplets. Using DIBs, the diffusion of membrane-permeable molecules and pore-mediated propagation of signaling molecules among droplets that recapitulate differentiation and simple feedback between sender and receiver droplets have been demonstrated (Figure 4C) [137]. Utilizing a similar design and a CFE system, Booth et al. created light-sensitive tissues made of droplets-in-oil that communicate only in the presence of external light triggers [143]. Even though αHL is the most common membrane protein in synthetic cell communication studies, other proteins such as MscL have also been used as a part of the signaling cascade or to mediate the propagation of signal molecules. For example, Haylock et al. have shown the communication of droplets-in-oil mediated by MscL G26C that opens upon external chemical stimuli [91]. In another work, Strutt et al. reconstituted MscL in DIBs, where MscL opening is triggered by membrane tension due to membrane asymmetry [92].

Membrane-less liquid–liquid phase-separated droplets can also be used as models of synthetic cells. Interactive behaviors such as prey and predator, for example, have been reconstructed between proteinosomes and coacervates [144]. Interactions between classic phospholipid-bound synthetic cells and hybrid synthetic cells or more uncommon coacervates that do not possess a physical boundary are open for exploration. Another potential platform for studying membrane–membrane interactions could be the recently discovered peptide bilayer for synthetic cell research [145,146].

### 5.2. Synthetic Cell–Natural Cell Communication

One of the pioneering works in the synthetic cell–natural cell communication was carried out by Lentini et al. [147]. In their work, the synthesis of αHL was controlled by a riboswitch that activated translation in the presence of a signaling molecule, theophylline. αHL then formed pores in the membrane of synthetic cells and allowed the release of IPTG, which, in turn, activated the synthesis of GFP in *E. coli*. Later, Lentini et al. designed synthetic cells that can sense the presence of *V. fischeri* through N-3-(oxohexanoyl)homoserine lactone (3OC6 HSL) and communicate with *E. coli* by synthesis of another homoserine lactone 3OC12 HSL, or participate in the *V. fischeri* quorum-sensing mechanism by synthesis and release of 3OC6 HSL (Figure 4D) [138]. Recently, the communication between cell-sized synthetic cells and bacteria was taken to a new level by engineering light-harvesting *E. coli* that creates proton gradients, leading to a pH change in the environment. By linking this pH change to pH-dependent DNA-origami attachment to the synthetic cell membrane, Jahnke et al. showed that synthetic cell shape change and deformation can be triggered by the proton pumping activity of *E. coli* [148]. Apart from compartmentalized synthetic cells, the DIB system has also been used to construct inducible gene circuits between *E. coli* and synthetic cells confined in droplets-in-oil [149].

Another intriguing yet more complicated form of synthetic cell–natural cell communication is the interaction between synthetic cells and eukaryotic cells. For example, synthetic enzymes have been compartmentalized in both liposomes and alginate microspheres to mimic the function of cytochrome P450 enzymes in dealkylation and hydroxylation of substrates. The products of these reactions then diffuse to mammalian HepG2 cells [150]. Even though the reaction products were fluorophore molecules, the work underscores visions to reconstitute more complicated synthetic cell–natural cell communication. In another work, Toparlak et al. constructed synthetic cells that contain or synthesize a neurotrophic factor that aids in neuronal differentiation and growth [151]. Most synthetic cells used in intercellular communication are based on small ~100 nm vesicles. The scarcity of work on cell-sized (~10 µm) synthetic cell–eukaryotic cell communication can be attributed to challenges including possible toxicity effects of synthetic cells, different timescales in synthetic cell life versus eukaryotic cell growth, and the stability of synthetic cells in physiological conditions.

Even though intercellular communication is a critical characteristic of living organisms and is responsible for their adaptability, growth, and survival, it is in its infancy for synthetic cells. The difficulty of reconstituting complex response and feedback systems to specific signaling molecules due to the limited pool of resources in a synthetic cell, non-specificity of membrane pores in allowing diffusion of molecules, and lack of transport mechanisms between synthetic cells create barriers for developing effective and efficient communication strategies between synthetic cells. This further illustrates the significance of reconstituting liposome fusion as it enables biomimicry of mechanisms found in exocytosis or viral infection. Efforts in mediating communication via more specific membrane proteins or fusion through specific DNA-pairing allow more specific targeted signal delivery and make efficient communication possible.

## 6. Summary

The desire to recreate complex cellular processes has led to the emergence of bottom-up synthetic biology. Synthetic cell research has propelled our understanding of biological processes, such as protein synthesis, exocytosis (membrane fusion), and cell-to-cell communications. We discussed different platforms of generating synthetic lipid bilayer membranes in the context of studying different ion channels and membrane proteins. There has been significant progress in generating giant vesicles with maximum encapsulation and minimum-to-no leakage [60], especially in droplet microfluidics [152].

Although DNA-mediated and coiled-coil peptide-mediated membrane fusion have gained popularity, they suffer from issues of controllability and stability of hemi-fusion or fusion intermediates. Recently, inter-cellular communications among synthetic cells and between synthetic cells and natural cells have received great attention in synthetic cell research [153]. We believe the next frontier of synthetic cell research will focus on developing increasingly sophisticated synthetic cell models that communicate with natural living cells.

## Figures and Tables

**Figure 1 membranes-11-00912-f001:**
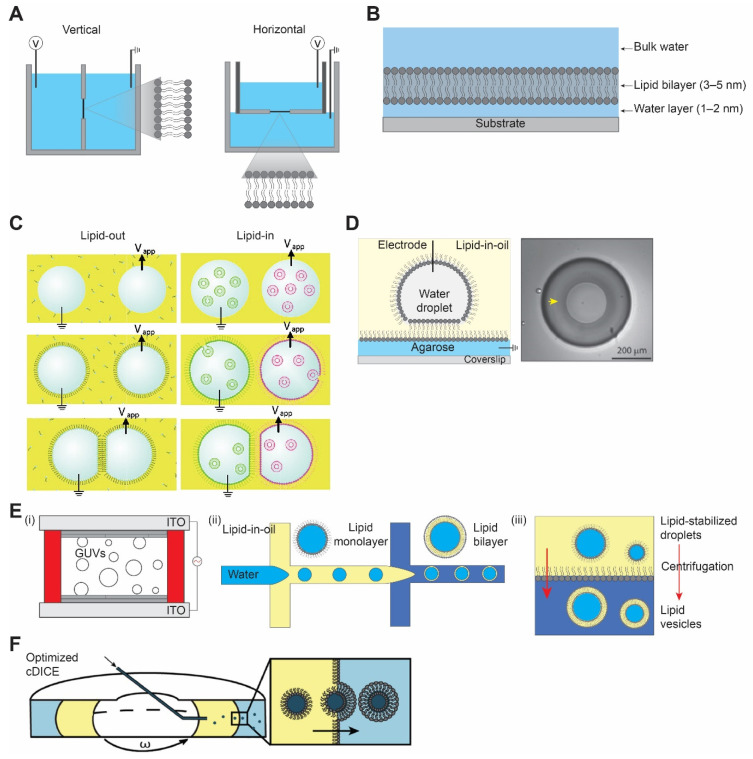
Different model bilayer platforms. (**A**) Black lipid membrane, vertical (left) horizontal (right). (**B**) Supported lipid bilayer membrane. (**C**) Schematic representation of droplet interface bilayer with lipid-in and lipid-out steps. Figures are reproduced from Reference [17] with permission from American Chemical Society, copyright 2008. (**D**) Droplet-hydrogel-supported bilayer on the left, and image of bilayer formed on the hydrogel surface (yellow arrow) on the right, top view. Adapted from Reference [18] with permission from American Chemical Society, copyright 2007. (**E**) Schematic illustration of (i) electroformation method, (ii) droplet microfluidics method, and (iii) inverted emulsion method for vesicle preparation. (**F**) Optimized cDICE cylindrical chamber and vesicle formation process by the centrifugal force. Reproduced from Reference [19] with permission from American Chemical Society, copyright 20212.1.2. Supported Lipid Bilayer.

**Figure 2 membranes-11-00912-f002:**
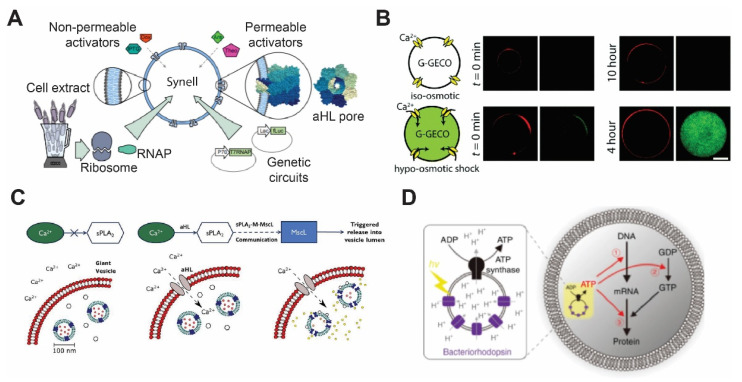
Membrane proteins’ incorporation in the lipid bilayer. (**A**) Diagrammatic representation of a synthetic cell with genetic circuit triggered by components that are transported from outside via αHL. Reproduced from Reference [81] with permission from Springer Nature, copyright 2016. (**B**) Construction of mechanosensitive synthetic cells expressing MscL. Under the hypo-osmotic condition, calcium ion penetrates through MscL and activates a genetically encoded calcium biosensor. Reproduced from Reference [83] with permission from The Royal Society of Chemistry, copyright 2017. (**C**) Vesicle-in-vesicle signaling pathway. Calcium influx occurring via αHL addition activates phospholipase A2 to trigger MscL to release quenched calcein dye molecules. Reproduced from Reference [84] with permission from National Academy of Sciences, copyright 2019. (**D**) Schematic illustration of artificial photosynthesis using the vesicle-in-vesicle approach and encapsulating bacteriorhodopsin and ATP synthase. Reproduced from Reference [85] with permission from Springer Nature, copyright 2019.

**Figure 3 membranes-11-00912-f003:**
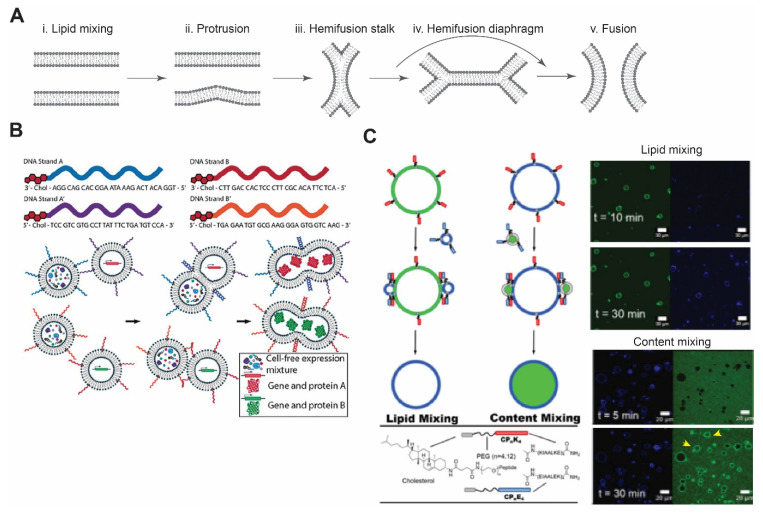
Membrane fusion. (**A**) Schematic of lipid membrane fusion showing sequential stages from protrusion to hemi-fusion and fusion pore. Redrawn from Reference [115] with permission from Springer Nature, copyright 2008. (**B**) Complementary DNA strands on two different vesicles eventually lead to their fusion and allow mixing of the contents. This can be utilized to initiate any biochemical reactions, such as in vitro protein synthesis. Reproduced from Reference [121] with permission from John Wiley and Sons, copyright 2019. (**C**) Schematic diagram of coiled-coil peptide-mediated vesicle fusion. Peptide K and peptide E were incorporated on the surface of vesicles using cholesterol with PEG as a linker. GUVs and LUVs were used in lipid and content mixing. Appearance of florescence signals after 30 min of incubation confirmed lipid mixing, while in content mixing, release of an encapsulated dye in the lumen of GUVs was observed (yellow arrows). Reproduced from Reference [122] with permission from Springer Nature, copyright 2020.

**Figure 4 membranes-11-00912-f004:**
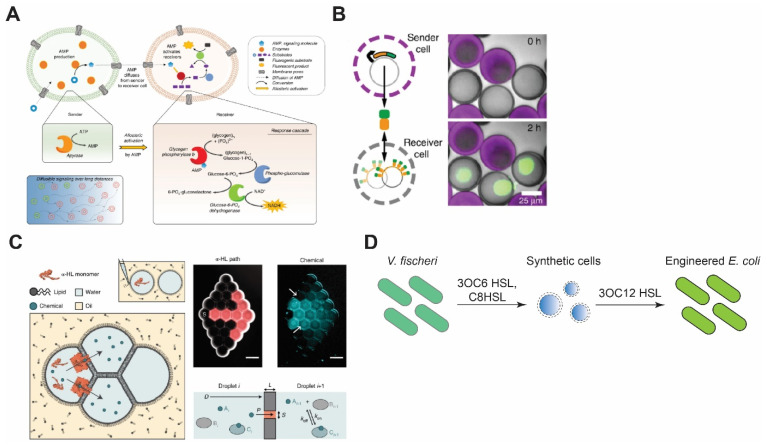
Synthetic cell communication with synthetic or living cells. (**A**) Signal amplification via allosteric activation of glycogen phosphorylase b by AMP. Sender cells generate AMP and send it through α-HL pores. Reproduced from Reference [135] with permission from Springer Nature, copyright 2020. (**B**) Porous synthetic cells that contain DNA-bound clay synthesize signal molecules and send them via chemical diffusion. Receiver cells encapsulate DNA sequences that encode binding sites for the signal molecules. Reproduced from Reference [136] with permission from Springer Nature, copyright 2018. (**C**) Reconstitution of synthetic cell communication in a network of droplet interface bilayers. The signal propagates through α-HL (left) or diffuses across the membranes (right) and activates cell-free expression of reporter proteins. Reproduced from Reference [137] with permission from Springer Nature, copyright 2019. (**D**) Synthetic cell–living cell communication via homoserine lactone molecules. Synthetic cells sense the presence of *V. fischeri* and send signal molecules to *E. coli*, thereby making *E. coli* sensitive to *V. fischeri* quorum-sensing molecules. Redrawn from Reference [138] with permission from American Chemical Society, copyright 2017.

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
