# Peer review of "Synthetic Cell as a Platform for Understanding Membrane-Membrane Interactions"

_membranes, 2021, doi:10.3390/membranes11120912_

Round 1

Reviewer 1 Report

Sharma et al. review recent progress in synthetic biology, focusing on the lipid platforms used to mimic different aspects of cell membrane behaviour. The review addresses fabrication methods (ranging from supported lipid bilayers to the transfer emulsion technique), highlights the incorporation of membrane fusion proteins and discusses intercellular communication between synthetic cells. Future research directions are mentioned, in particular those dealing with interactions between natural cells and artificial cells. 
The review is well-written, it is timely, and it addresses the readership of Membranes. I would recommend publication after considering the minor points below: 
-The authors could mention that many of the GUV systems they review are also interesting as protocells, i.e.  as hypothetical precursor structures of the first cells in origin of life studies. 
-The authors could expand their discussion on DNA to include incorporation of DNA nanostructures (DNA origami pores) as synthetic alternatives complementing the ones presented in section 3.  They already mention such an example in Ref [128].
- Would be good to also discuss the drawbacks of using lipid vesicles. The stability of the vesicles described in the review makes polymersomes an interesting alternative to lipid vesicles. The authors briefly mention such structures (page 8, 306), but do not introduce polymersomes to the reader.  

Author Response

Response to Reviewers

We thank the reviewers for their critical evaluation and suggestions. We also thank for their positive comments. We have provided a point-by-point response below. The original comments from the reviewers are in black text while our responses are in blue and the suggested changes in the manuscript are in red.

Referee: 1

Sharma et al. review recent progress in synthetic biology, focusing on the lipid platforms used to mimic different aspects of cell membrane behaviour. The review addresses fabrication methods (ranging from supported lipid bilayers to the transfer emulsion technique), highlights the incorporation of membrane fusion proteins and discusses intercellular communication between synthetic cells. Future research directions are mentioned, in particular those dealing with interactions between natural cells and artificial cells. 
The review is well-written, it is timely, and it addresses the readership of Membranes. I would recommend publication after considering the minor points below:

  1. The authors could mention that many of the GUV systems they review are also interesting as protocells, i.e.  as hypothetical precursor structures of the first cells in origin of life studies.

We thank the reviewer for the comment. We have included the following text in the manuscript on page 4, line 160.

Giant unilamellar vesicles (GUVs) are the most widely used model as synthetic cells and as protocells from the origin of life perspective [40].

  1. The authors could expand their discussion on DNA to include incorporation of DNA nanostructures (DNA origami pores) as synthetic alternatives complementing the ones presented in section 3.  They already mention such an example in Ref [128].

We thank the reviewer for the comment. We have included the following text in the manuscript on page 6, line 223.

Inspired by biological nanopores such as αHL, there has been significant advancement in the field of synthetic nanopore, especially ones designed with DNA [70,71]. Unlike αHL, the size of DNA-based nanopore can be tuned from 4 to 30 nm in diameter [72–74] and these nanopores have been applied in numerous sensing applications [75].

  1. Would be good to also discuss the drawbacks of using lipid vesicles. The stability of the vesicles described in the review makes polymersomes an interesting alternative to lipid vesicles. The authors briefly mention such structures (page 8, 306), but do not introduce polymersomes to the reader.

We thank the reviewer for the comment. We have introduced polymersomes in the vesicle preparation section in the manuscript on page 4, line 162.

Giant vesicles made of lipids are fragile and this has motivated the use of other membrane-forming materials. Polymersomes are comprised of chemically synthesized amphiphilic polymers that self-assemble to form vesicles [41]. They are robust and considered to be mechanically tough with low-permeability for ions as compared to lipid vesicles. Here we focus our discussion on lipid bilayer giant vesicles as lipids are the natural substrate for membrane proteins.

Reviewer 2 Report

The authors present a review regarding  the highly interesting topic of synthetic cells as a platform for understanding membrane interactions.

However, the main body of the manuscript refers to (synthetic) membrane architectures and only at the end, the highly interesting part regarding synthetic-cell communication appears. 

In the membrane - method part, actually, some aspects are misleading/missed out, e.g. not defined precisely - the use of lipids as building blocks of membranes is often used in the the context of self-assembly and not named 'synthetic membranes'. 

The allover citations in this review appear at least partially, superficial, as they do refer often to review articles and not to the original work.

In detail: in chapter 1, what does the authors mean by 'unique setup' in line 67?

and in line 80, the presence of 'extra solvent'? the term 'direct insertion' in line 88 is highly misleading, as this process is termed reconstitution by 'undergoing the critical micelles concentration', which (the presence of detergent) is also an often underestimated parameter in a-HL (reconstitution).

in the chapter supported lipid mailers, the hydration is mentioned as interacting force, which - at least - needs to be elaborated, otherwise, it is highly misleading. 

the chapter regarding solid supported lipid membranes is missing and the motivation for droplet interface bilayers is contradicting the motivation for the solid supported membranes, namely: mechanical stability (which is, of course, a very relative feature of lipid-based membranes in general)

the research regarding synthetic (polymeric) membranes is not described in sufficient details, even though, this could solve the issue of mechanical stability.

Also the in vitro synthesis method appears as a quite young field and the authors are not reflecting the history of the field, e.g the  identification and development of cell free synthesis, membrane protein reconstitution (Nirenberg and Matthei et al.  Swartz et al., Ueda et al....) in spherical lipid systems, and,  in tethered planar lipid membranes, e.g. Robelek et al. and (sphaerical) polymeric membranes, e.g. Discher et al, May, et al..  

The following chapters appears as a list without reasoning the connection of the methods to the main topic: membrane interaction. It would be helpful, if advantages/disadvantages of the listed methods towards the major topic, membrane interaction, would be guiding the reader through this review article. 

In summa, the review tackles a very interesting and upcoming field: understanding membrane interactions by use of synthetic cells - however the main body appears not related to this topic, or even misleading as the process of insertion by the ribosomal complex is not elaborated, but methodological aspects, such as the setup of BLM characterisation. In this stage, the manuscript appears superficial and unfocused. 

Author Response

Response to Reviewers

We thank the reviewer for their critical evaluation and suggestions. We also thank for their positive comments. We have provided a point-by-point response below. The original comments from the reviewers are in black text while our responses are in blue and the suggested changes in the manuscript are in red.

REVIEWER REPORT(S):

Referee: 2

The authors present a review regarding the highly interesting topic of synthetic cells as a platform for understanding membrane interactions.

However, the main body of the manuscript refers to (synthetic) membrane architectures and only at the end, the highly interesting part regarding synthetic-cell communication appears.

  1. In the membrane - method part, actually, some aspects are misleading/missed out, e.g., not defined precisely - the use of lipids as building blocks of membranes is often used in the context of self-assembly and not named 'synthetic membranes'. 

We thank the reviewer for the comment and apologize for the confusion. We have removed all mentioning of synthetic membranes in the manuscript and refer to them as lipid bilayer membrane.

  1. The overall citations in this review appear at least partially, superficial, as they do refer often to review articles and not to the original work.

We thank the reviewer for the comment. We have now cited more original work and important review articles to provide the readers with a more comprehensive perspective of the literature.

  1. In detail: in chapter 1, what does the authors mean by 'unique setup' in line 67? and in line 80, the presence of 'extra solvent'? the term 'direct insertion' in line 88 is highly misleading, as this process is termed reconstitution by 'undergoing the critical micelles concentration', which (the presence of detergent) is also an often underestimated parameter in a-HL (reconstitution).

We thank the reviewer for these questions. By saying BLM is a ‘unique setup’, we want to emphasize the fact that when researchers designed this setup back in 1960s, it was ‘unique’ in of itself where both the aqueous chambers were accessible to use, with an electrode in each chamber.

In line 80, we stated

Incorporation of membrane proteins to BLM is usually achieved by direct insertion of purified proteins into the bilayer or incorporated by fusion of proteoliposomes to the standing bilayer membrane.’

‘Insertion’ is very well accepted description for reconstitution of membrane proteins. (Insertion and folding pathways of single membrane proteins guided by translocases and insertases, PMID: 30801000, Membrane protein insertion: mixing eukaryotic and prokaryotic concepts, PMID: 16264426)

By direct insertion, we want to state that insertion of purified membrane protein can occur without the need of any assistance like detergent. This statement is in the reference to insertion of α-hemolysin. We further stated in the same sentence that incorporation of membrane proteins can also be achieved by proteoliposomes fusing with the BLM. The reason for not including micelles/detergent-assisted insertion of membrane proteins in BLM is that it could solubilize the standing bilayer membrane.

  1. In the chapter supported lipid bilayers, the hydration is mentioned as interacting force, which - at least - needs to be elaborated, otherwise, it is highly misleading.

It is not clear to us what is misleading about hydration force. The interactions between a planar bilayer and its underlying solid support are controlled by van der Waals, electrostatic, hydration, and steric forces. Strongly hydrated surfaces exert a repulsion force between them because of the energy required for perturbation of the ordered structure of the bound layers of water molecules. Hydration forces is also the reason why liposomes of certain compositions do not aggregate in the presence of multivalent cations. We included references to support the statement and added (a repulsion force between two hydrated surfaces) after hydration.

  1. The chapter regarding solid supported lipid membranes is missing and the motivation for droplet interface bilayers is contradicting the motivation for the solid supported membranes, namely: mechanical stability (which is, of course, a very relative feature of lipid-based membranes in general).

We thank the reviewer for the comment. There is a sub-section 2.1.2 on supported lipid bilayer on page 3. The motivation for developing droplet interface bilayer was to minimize the volume of the aqueous phase and provide accessibility to add reaction components from either side of the lipid bilayer.

  1. The research regarding synthetic (polymeric) membranes is not described in sufficient details, even though, this could solve the issue of mechanical stability.

Thank you for the comment. We have briefly mentioned polymersomes in the vesicle preparation section on page 4.

  1. Also the in vitro synthesis method appears as a quite young field and the authors are not reflecting the history of the field, e.g the identification and development of cell free synthesis, membrane protein reconstitution (Nirenberg and Matthei et al.  Swartz et al., Ueda et al....) in spherical lipid systems, and, in tethered planar lipid membranes, e.g. Robelek et al. and (spherical) polymeric membranes, e.g. Discher et al, May, et al.

We thank the reviewer for this comment. In vitro protein synthesis indeed has been established since the early days of biochemistry. Their use in conjunction with model membranes, including in the context of synthetic cells, is a more recent focus of the field. We have now cited several suggested articles in the manuscript.

  1. The following chapters appears as a list without reasoning the connection of the methods to the main topic: membrane interaction. It would be helpful, if advantages/disadvantages of the listed methods towards the major topic, membrane interaction, would be guiding the reader through this review article. 

We thank the reviewer for the comment. We have modified the title of the manuscript to emphasize membrane-membrane interactions, realizing that membrane interaction may have given the wrong impression of our review.

We have discussed the disadvantages and advantages of methods/studies throughout the manuscript. Without specific examples provided by the reviewer, we are unsure where exactly changes are needed.

We have added a few sentences in different sections to make a connection with sub-headings and methods described in the review.

Page 2, line 91:

BLM is most useful for probing the biophysical properties of channels. For example, if a membrane protein has a conformational change that induces downstream effects, such effects will be difficult to detect in a BLM setup. Hence, utilizing this platform is mostly beneficial when studying the selectivity, conductivity, and drug pharmacology properties of reconstituted proteins.

Page 4, line 148:

Due to the defined compartmentalization in this method, DIBs have been used to recapitulate communication pathways and feedback cascades mediated by membrane proteins and porins. The ability to visualize diffusion of fluorescent molecules across droplets as well as to record electrical currents between droplets makes DIBs ideally suited for studying protein-mediated interactions between droplet-in-oil synthetic cells. The downside, however, is that the outer organic phase does not resemble the aqueous solutions in a natural environment and an oil-water interface may result in protein denaturation.

Page 6, line 231:

The majority of interactions that membranous structures have with either each other or their external environment are mediated by membrane proteins. Using synthetic cells as model membranes for studying membrane-membrane interaction is not possible unless their membrane is decorated with functional proteins that allow interactions with outer environment.

Page 11, line 428:

Membrane proteins such as GPCRs are the critical in signaling cascades for cells to respond to changes in their environment. The advances in membrane protein reconstitution methods described earlier have led to significant progress in reconstituting intercellular communication among synthetic cells.

In summary, the review tackles a very interesting and upcoming field: understanding membrane interactions by use of synthetic cells - however the main body appears not related to this topic, or even misleading as the process of insertion by the ribosomal complex is not elaborated, but methodological aspects, such as the setup of BLM characterisation. In this stage, the manuscript appears superficial and unfocused. 

We thank the reviewer for recognizing this is an interesting topic and for the critique. We understand that there may be elements of the review that the reviewer may disagree with. However, we have a particular focus that we wish to bring out and believe we have done that in the present text. The take home message is to highlight the use of synthetic cells as a model system for studying membrane-membrane interaction.

Round 2

Reviewer 2 Report

The authors have reworked the review article at some crucial points, it is a comprehensive and well structured review article about this very relevant and emerging field.